# Experiences of supported isolation in returning travellers during the early COVID-19 response: a qualitative interview study

Holly Carter ![ORCID] ,[1] Dale Weston,[1] Neil Greenberg,[2] Isabel Oliver,[3] Charlotte Robin,[1] G James Rubin,[2] Simon Wessely ![ORCID] ,[2] Louis Gauntlett,[1] Richard Amlot[1]

[1]Behavioural Science and Insights Unit, Public Health England, Porton Down, Salisbury, UK
[2]Department of Psychological Medicine, King's College London, Weston Education Centre, London, UK
[3]National Infection Service, Public Health England, Bristol, UK

**Correspondence to**
Dr Holly Carter;
holly.carter@phe.gov.uk

## ABSTRACT

**Objectives** (1) To understand the experiences and perceptions of those who underwent supported isolation, particularly in relation to factors that were associated with improved compliance and well-being; (2) to inform recommendations for the management of similar supported isolation procedures.

**Design** We carried out a qualitative study using semistructured interviews to capture participants' experiences and perceptions of supported isolation. Data were analysed using the framework approach, a type of thematic analysis that is commonly used in research that has implications for policy.

**Setting** Telephone interviews carried out within approximately 1 month of an individual leaving supported isolation.

**Participants** 26 people who underwent supported isolation at either Arrowe Park Hospital (n=18) or Kents Hill Park Conference Centre (n=8) after being repatriated from Wuhan in January to February 2020.

**Results** Six key themes were identified: factors affecting compliance with supported isolation; risk perceptions around catching COVID-19; management of supported isolation; communication with those outside supported isolation; relationship with others in supported isolation; and feelings on leaving supported isolation. Participants were willing to undergo supported isolation because they understood that it would protect themselves and others. Positive treatment by staff was fundamental to participants' willingness to comply with isolation procedures. Despite the high level of compliance, participants expressed some uncertainty about what the process would involve.

**Conclusions** As hotel quarantine is introduced across the UK for international arrivals, our findings suggest that those in charge should: communicate effectively before, during and after quarantine, emphasising why quarantine is important and how it will protect others; avoid coercion if possible and focus on supporting and promoting voluntary compliance; facilitate shared social experiences for those in quarantine; and ensure all necessary supplies are provided. Doing so is likely to increase adherence and reduce any negative effects on well-being.

## INTRODUCTION

The first cases of a novel strain of coronavirus (SARS-CoV-2) were detected in Wuhan,

## Strengths and limitations of this study

► To our knowledge, the present study is the first research conducted with individuals during and immediately following their supported isolation in the UK as part of the COVID-19 response.
► We used semistructured interviews to gain an indepth understanding of the experiences of a sample of people (n=26) who underwent supported isolation.
► Interviews were carried out within 1 month of participants leaving supported isolation.
► Our findings are highly topical given the recent introduction of a requirement for travellers to the UK to isolate within hotel accommodation.
► It was not possible to interview everyone who underwent supported isolation, and we were only able to interview those who had a good understanding of English.

China, in December 2019. On 31 January 2020, British nationals living in Wuhan were offered repatriation to the UK. Ninety-three returned on two chartered flights. In order to be repatriated, all had to agree to undergo 14 days of 'supported isolation'. In some countries and contexts, this type of supported isolation is known as quarantine; however, it is typically referred to as supported isolation in the UK, and so will be referred to as supported isolation in the current study. Supported isolation took place in an accommodation block at Arrowe Park Hospital in the Wirral,[1] and Kents Hill Park Conference Centre, Milton Keynes. All supported isolation ended by 23 February 2020.[2] On arrival at the supported isolation facility, individuals were provided with their own rooms which were fully furnished and had basic cooking, washing and living facilities.[3] Individuals were encouraged to stay in their rooms as much as possible (though this was not mandatory) and could access anything they needed by

phoning staff or using an online system; if they did need to leave their rooms they were encouraged to follow hand hygiene guidance and wear a face mask. Individuals also had access to a team of medical staff who closely monitored their condition, including regular testing and symptom checking.[3] There was phone and internet access to enable them to communicate with others both inside and outside the supported isolation facility.

Many countries, including China,[4] Vietnam[5] and Singapore,[6] have had supported isolation policies in place in response to COVID-19 for over a year, for a variety of situations including international travel. However, supported isolation for returning travellers had, to our knowledge, never been used before within the UK. It was anticipated that the experience could have considerable psychological consequences for the individuals concerned, including potential post-traumatic stress, anger and confusion; consequences that may be affected by a range of stressors including information provision, stigma and fear of infection.[7] Furthermore, supported isolation represents a unique social context in which relative strangers are placed in close quarters within a novel context and asked to adhere to recommended behaviours for a prolonged period. During emergencies, such social contexts can affect individuals' social identity, which can have consequences for adherence and psychological resilience.[8–10] Outside of the emergency response context, the emergence of strong social connections among strangers in close physical proximity has been associated with positive well-being-related outcomes.[11]

From 15 February 2021, those travelling to the UK from 'red list' countries (countries which have higher prevalence of new COVID-19 variants)[12] have been required to isolate in hotels for 10 days.[13] Countries on the 'red list' are continually reviewed and updated, but as of 9 April 2021 there were 39 countries on the list.[14] Policy around this isolation is focused on identifying the best ways to maximise compliance, with an increasing emphasis on enforcement.[15] Furthermore, with the COVID-19 pandemic ongoing, it is possible that supported isolation will be required in other contexts, such as to assist those with difficulty isolating at home[16] or to reduce household transmission.[17] It is therefore important to understand more about the way in which people experience supported isolation, so that this process can be optimised to increase adherence and mitigate any negative effects on well-being. We carried out a rapid mixed-methods study in which we: (1) interviewed individuals who underwent supported isolation at Arrowe Park Hospital and Kents Hill Park Conference Centre (findings reported here); (2) surveyed those who underwent supported isolation at two time points (immediately after supported isolation and 3 months after supported isolation) (findings to be reported elsewhere). To our knowledge, this is the first research conducted with individuals during and immediately following their supported isolation in this country. With supported isolation now being required for people travelling to the UK from a number of countries,

the findings presented here will be invaluable in understanding public experiences of supported isolation and informing optimised management in these settings.

## Aims

This study had two aims: (1) to understand the experiences and perceptions of those who underwent supported isolation, particularly in relation to factors that were associated with improved compliance and well-being; (2) to inform the development of recommendations for the management of similar supported isolation procedures.

## METHOD

### Patient and public involvement

Given the extremely rapid and responsive nature of this research, it was not possible to involve patients or the public in the development of the study and associated materials. However, staff at the supported isolation facilities were involved from the outset in planning the study and facilitating participant recruitment. Additionally, findings from this study will be shared with participants on publication.

### Design

This study used semistructured interviews to capture participants' experiences and perceptions of supported isolation. The decision was taken to carry out semistructured interviews (alongside surveys, findings to be reported elsewhere) in order to generate a more in-depth understanding of participants' perceptions and experiences during supported isolation than could be obtained using surveys alone. Telephone interviews took place within 1 month after the isolation. The study was designed and carried out in line with Consolidated Criteria for Reporting Qualitative Research guidelines[18] (see online supplemental appendix 1).

### Participants

Participants underwent supported isolation in either Arrowe Park (n=18) or Kents Hill Park (n=8) in January and February 2020. The day before leaving supported isolation, all those in the supported isolation facilities were provided with an information sheet about the study by a member of staff at the facility. This included an invitation to take part in a survey (findings to be reported elsewhere), as well as the opportunity to take part in an interview. Thus, voluntary response sampling was used, whereby all those who underwent supported isolation were given the opportunity to take part in both the survey (findings to be reported elsewhere) and an interview, and the sample consisted of those who chose to opt in to the study. To opt in to the interview part of the study, participants were asked to provide an email address on leaving supported isolation to enable the research team to follow-up and arrange the interview. At this point, 69 people provided a contact email address, and all were then contacted separately and invited to take part in an

interview. Of these, 26 people (38%) consented to take part in an interview; this sample therefore represents 12.3% of the entire population who underwent supported isolation.

## Materials

An interview schedule was developed to capture in-depth information about individuals' experiences and perceptions of supported isolation, including their: overall experience (eg, 'Tell me about your experience of undergoing supported isolation'); willingness to undergo supported isolation (eg, 'Were you willing to undergo supported isolation?'); perceptions of the way the supported isolation process was managed (eg, 'In general, how do you feel the supported isolation process was managed?'); perceptions of others' behaviour during supported isolation (eg, 'How did those in supported isolation behave towards each other?'); experiences after leaving supported isolation (eg, 'How has life been for you since leaving supported isolation?'). See online supplemental appendix 2 for a copy of the interview schedule.

## Procedure

Each interview took place within 1 month of leaving the supported isolation facility and lasted for approximately an hour. Interviews were carried out by behavioural scientists based at Public Health England (PHE) or King's College London, all of whom were qualified to at least MSc level and had received training in carrying out interviews. Researchers did not establish a relationship with participants prior to carrying out the interview nor were participants made aware of any personal characteristics of the interviewer, aside from their place of work and the broad aims of the research. Interviews were carried out by both male and female members of the research team. Only the researcher and the participant were present during the interview. Prior to taking part in an interview, participants completed a written consent form. They also provided verbal consent at the start of the interview. Interviews were recorded and subsequently transcribed. After taking part in an interview, participants received a debriefing statement which provided further information about the study, as well as sources of support that participants could access if required.

## Analysis

All interviews were completed before beginning data analysis, at which point a framework approach was used to analyse the data.[19] This is a type of thematic analysis that is commonly used within research that has implications for policy and practice.[20] After familiarisation with the data, an initial coding framework was developed based largely on a priori areas of interest in line with the research aims, and specifically included factors that have been shown during previous incidents to be related to compliance and well-being.

At this stage, themes were also allowed to emerge from the data. The initial coding framework was intentionally

**Table 1** Description of themes and subthemes

| Theme | Subtheme |
|---|---|
| Factors affecting compliance | Factors promoting compliance |
| | Factors threatening compliance |
| Risk perceptions around catching COVID-19 | Low perceived risk |
| | High perceived risk |
| Management of supported isolation | Operational management |
| | Treatment by staff |
| | Communication from staff |
| Communication with those outside | |
| Relationship with others within supported isolation | |
| Feelings after leaving supported isolation | |

broad, to ensure that areas of interest were not missed, and contained a total of 76 categories, within 22 major themes. The initial framework was discussed with a second researcher, who had also familiarised themselves with the data, and then applied to a small number of transcripts. The initial coding framework was then refined into an analytical framework, in which codes were grouped together into overarching themes. This resulted in six key themes and seven subthemes. See table 1 for a full breakdown of themes and subthemes.

Application of the analytical framework was carried out by hand by the first author, with each passage in the data being coded into one or more of the identified themes. A spreadsheet was used to generate a matrix into which relevant data (eg, passages of interest relating to each theme) were organised thematically. This enabled data to be compared and contrasted within and between themes and facilitated more in-depth interpretation. After analysing the 26 transcripts no new themes emerged, thus data saturation had been reached.[21]

## RESULTS

### Demographics

Half of the participants (n=13) were male and half (n=13) were female. Participants ranged in age from 22 to 78 (mean=43.2 years). The majority of participants were British nationals (n=22), with a small number of Chinese nationals (n=3) and one person who selected 'Other' as their nationality. Similarly, the majority of participants were White British (n=17), or Chinese (n=7), with one person being White Irish, and another being Black British. Most participants were educated to degree level or above (n=17), with a smaller number being educated to higher secondary level (n=8), and one being educated to primary or lower secondary level. The majority of participants were employed either full time

(n=14) or part time (n=4). A small number were retired (n=4), unemployed (n=2) or self-employed (n=1), with one participant specifying that they were due to start work following their isolation.

Participants were asked what their reason was for being in Wuhan during the COVID-19 outbreak, and most stated that they were either living there (n=6), visiting family or friends (n=8) or on holiday (n=5). A smaller number were there on a business trip (n=2), with one participant having been deployed as part of the Foreign and Commonwealth Office (FCO) response. A small number stated that they had not been in Wuhan and were isolating on their return from other affected areas, including Hubei province (n=2) and the Diamond Princess cruise ship (n=2). The majority of participants were travelling either with family (n=11) or on their own (n=10), with a small number travelling with others they had no relationship with (n=5). The majority of participants did not share a room (n=15). Of those who did (n=11), most shared with family (n=7) or friends (n=1), with only a small number sharing with people they did not know (n=3).

### Factors affecting compliance with supported isolation
#### Factors promoting compliance
Most participants were willing to undergo supported isolation. They understood why supported isolation was necessary and why they were being asked to undergo it, for example, 'I understood the necessity and I was willing to cooperate very much' (KHP2). Most participants felt that the positives of supported isolation outweighed the negatives. Positive aspects were grouped broadly into three themes: a belief that supported isolation protects family and friends as well as UK society, for example, 'it was in our best interests and the people we love in the UK and the country in general' (KHP8); a belief that supported isolation would protect themselves, by ensuring they were in a safe place if they developed symptoms and that they would not be blamed in the event of an outbreak in the UK, for example, 'in the event that I or any of my fellow travellers developed symptoms we would be in that hospital environment or we would be with doctors who spoke our native language' (AP9); and faith in the effective management of the supported isolation process, for example, 'when we actually arrived at Arrowe Park […] the staff there gave such a warm welcome and made everything feel so sort of warm and comfortable' (AP16).

#### Factors threatening compliance
Where participants expressed concerns these centred around uncertainty about what the process would involve, for example, 'You're thinking well what are the facilities here going to be like? How am I going to cope with that?' (AP24), sometimes attributing this to lack of information being provided, for example, 'I was a little bit apprehensive just because I didn't know […] how it would be structured or organised, and obviously the lack of details' (AP11). Others were concerned that they would

be bored, for example, '[I was concerned that] I would be a bit bored' (AP19) or would be at increased risk of catching COVID-19, for example, 'Our biggest concern would be is anybody sick because of this virus among us?' (KHP2).

A few felt angry or frustrated about the process because they did not think it was necessary, for example, 'we did think it was unnecessary because we were already tested negative' (AP24) or believed it was a waste of time and resources, for example, 'it was an over the top response that probably cost 2 or 3 million pounds for those two weeks' (KHP4). In the few instances where participants did not want to comply, non-compliance took the form of breaking the rules inside the supported isolation facility (eg, trying to obtain more alcohol than was allowed), but not trying to leave the supported isolation facility, for example, 'Over a short period of time it was let's try and break the rules just for something to do. Let's see how far we can go' (KHP4).

### Risk perceptions around catching COVID-19
#### Low perceived risk
Participants reported different perceived risks of catching COVID-19 while in isolation. Some felt at low risk because they could take protective behaviours, for example, 'we were just very careful with washing our hands […] just sensible hygiene precautions really. So that made us feel pretty safe' (AP16). The most commonly reported protective behaviours included staying in their own room, for example, 'we just decided not to go out, just to stay in our hotel rooms' (KHP2), observing effective hand hygiene, for example, 'I would wash my hands when I went downstairs' (AP11), and wearing a face mask, for example, 'we were wearing gloves and masks and keeping no more contact with each other' (KHP3). Other reasons given for low perceptions of risk included that anyone displaying symptoms could be quickly isolated, for example, 'I knew that things were being monitored very carefully and things were being done about it' (KHP4), and that everyone in the supported isolation facility underwent regular testing, for example, 'after one week we'd all been tested negative, after 10 days we'd all been tested negative, after 14 days we'd all been tested negative' (KHP4).

#### High perceived risk
However, others were very worried about catching COVID-19 during their stay in supported isolation. Common reasons for this included other people having symptoms, for example, 'someone with a high temperature, she was really close to me, so I said oh please don't stay too close' (AP10), and the need to sometimes be in close proximity to others, for example, 'we were using the same big meeting room for one or two hours before we eventually went to our separate rooms' (KHP2). However, most participants stated that their risk perception reduced over time in the facility, as people continued to test negative, and did not have any symptoms, for example, 'towards the end of the isolation, it was getting clearer

that nobody in there was probably carrying the virus […] you didn't feel like there was a threat of catching anything from anybody' (AP16). The majority of participants noted that they felt most worried at the start of the supported isolation process.

## Management of supported isolation
### Operational management
Most participants reported that they felt the whole process was well managed. Reasons for this included that the process was well organised, for example, 'the place all sort of ran like clockwork from my point of view' (AP16), and that staff and management were willing to adapt procedures following negative feedback about the process, for example, 'the food initially it was only microwave meals but that evolved in the second week […] everybody was sort of learning as we went along' (KHP5).

Where participants did express concerns these often centred on provision of food, including: not receiving meals, for example, 'they forgot to give me breakfast and lunch three times' (KHP1); food being served uncovered, for example, 'I think most of us had the salad or the bread which was not covered' (KHP2); and food not being warm enough, for example, 'the food turned up lukewarm in cardboard boxes' (AP22). Relatedly, several participants felt that the cultural background of those undergoing supported isolation had not been properly considered. Many travellers were Chinese nationals and fresh food is very important to people in China, for example, 'when they chose a facility that didn't have fresh food on site they didn't understand the Chinese way of life' (KHP8); the ready meals and pre-prepared foods provided in the first few days of supported isolation were therefore inappropriate.

Another area of management that participants suggested could be improved was around internal communication within and between organisations, for example, 'With the change of shifts, they didn't update people […] there was no passing on of communication, there was no register of requests from room numbers' (AP18). A final consideration raised in relation to operational management of supported isolation was that several participants would have liked more access to outside space and exercise facilities, for example, 'outdoor space improvements may have been helpful […] I think we are all finding value in still being able to get outside a little bit' (AP12). For the most part, participants who provided negative feedback about the operational management of the supported isolation process felt that changes were made to address their concerns, and that the management of the supported isolation process improved as time went on, for example, 'they are improving their responding and they are learning from their mistakes as well they were really good I was really impressed' (KHP6).

### Treatment by supported isolation staff and authorities
Overall, participants were extremely positive in their feedback about the way in which staff treated them. The staff

were friendly and helpful, for example, 'we were treated with compassion […] and so we were immediately put at ease' (AP24), went out of their way to keep people happy, for example, 'the staff went above and beyond in trying to help us' (AP11), and provided people with anything that they asked for, for example, 'staff were very helpful, whatever we asked they tried to answer, and whatever we needed they tried to procure' (AP13). A few participants mentioned that staff did not try to avoid them or treat them as if they were ill, for example, 'we don't feel that really we were isolated or we were frightening […] as somebody who might carry a virus' (KHP2). A small number of participants specifically noted that staff achieved a good balance between promoting good public health, without making the process too restrictive, for example, 'I think that's a balance that had to be struck between health risk and […] how we felt that we were being treated, how restricted we felt' (AP11).

### Communication from staff during supported isolation
Participants were also overwhelmingly positive about the way in which members of staff communicated with them. Almost all participants talked about the daily newsletter that they received from staff and felt that this was an effective way of providing information about protective actions, timings of any activities and testing, for example, 'I think they were really good…we would get two or three letters a day actually sometimes about what was changing and why' (AP13). Similarly, participants noted that staff were proactive in their communications, calling regularly to check on each individual, for example, 'in the mornings when a nurse would come around […] if there were any sorts of developments to tell us about then they would' (AP15), and scheduling regular update meetings. Participants also felt that staff answered all their questions (or tried to) and were open and transparent in providing information, for example, 'I think they would have answered anything that we needed to know' (AP22).

Some expressed dissatisfaction at the somewhat old-fashioned methods of communication, for example, 'their way of disseminating information was posting things under the door, which […] seems a little old-fashioned […] maybe if they had done a group chat or done a group email […] I think that may have been a good way of communicating' (AP16), and information not being provided in multiple languages, for example, 'the Mum […] had to ask for a lot of help because of her difficulties with English, she was a Chinese national' (AP11). A small number of participants also felt that staff had been unable to answer some questions, for example, 'the only information [that staff couldn't give me] was sort of about leaving actually, and what was going to happen […] that information was only very near the end' (AP15).

### Communication with those outside of supported isolation facilities
Most participants found it easy to communicate with those outside supported isolation and did so regularly. Several

participants expressed how important this was in helping them to get through the supported isolation process, for example, 'we spent half the day usually emailing and skyping and WhatsApping everybody […] it was actually good having that routine' (AP24). Additionally, some were able to carry on working during supported isolation, and this helped them to pass the time. A few participants also highlighted the benefit of local community groups who posted pictures of uplifting things, for example, 'it's nice when you are in that situation […] to see stuff that wasn't about the virus, and wasn't doom and gloom' (AP25).

On the other hand, some participants did note difficulties in communicating with those outside of supported isolation, and these typically related to having limited access to internet or poor phone signal, for example, 'the phone signal where we were was terrible' (AP22).

### Relationship with others within supported isolation

Where people felt a connection with others this was often due to a sense of camaraderie, for example, 'I think there was a bit of camaraderie […] everyone was in the same situation really' (AP16), or shared experience, for example, 'we were all in the same boat […] it was just, we were all in it together really' (AP22). Some participants described how people supported and encouraged each other during the supported isolation process, for example, 'we look after each other, we tried to be helpful with each other as well' (KHP3), stating that this helped people to get through the experience, for example, 'we encouraged each other and things like that sometimes. It was good to help many to spend the long and sometimes worrying days' (KHP2). This connection was facilitated by the formation of chat groups, for example, 'we would message on Facebook and WhatsApp and all that stuff' (AP23), and some level of freedom to socialise with others, for example, 'we had a little common room within our side of the conference centre […] so we did movie nights and quizzes and things like that' (KHP5).

Where people did not feel a connection with others this was because they either did not get the opportunity to interact much with others or actively avoided it (due to fears about catching COVID-19), for example, 'they all got together and things like that and the invitation was open but at the same time I didn't really want to be in the same room with lots of people' (AP13).

Most participants felt that they could trust others to behave appropriately and instances of uncooperative behaviour were rare or non-existent, for example, 'people were very very well-behaved […] people are grateful that was a common feeling' (KHP6). A handful of participants noted isolated instances of uncooperative behaviour, for example, 'there's only one argument that we ever heard in the whole two weeks and it was somebody saying that they've been tested negative three times can they go home early […] but apart from that the whole two weeks was like with no issue at all' (KHP8), but almost all said that the majority of people were friendly and cooperative, for example, 'Almost all […] were quite cooperative […] I think they were quite friendly to each other' (KHP2).

### Feelings after leaving supported isolation

Many participants felt happy and relieved to leave supported isolation and get back to normal, for example, 'I've never been so happy to see my own bed […] and my own house' (AP16). Most people felt that others had treated them normally on leaving supported isolation, and that they had not experienced negative reactions from others, for example, 'nobody has reacted any different to me' (AP17).

However, several participants stated that they struggled after leaving supported isolation. Some felt anxious or overwhelmed, with reasons including not being used to going outside, for example, 'I actually had a panic attack when I got in the taxi I found everything very overwhelming […] I hadn't really mentally prepared myself for going outside' (AP23), or being concerned about mixing with large numbers of people again, for example, 'First time we went to the supermarket […] just seeing people who were not in masks and protective clothing took some getting used to […] all the crowds of people in the supermarket when we'd just been used to us two was quite uncomfortable' (AP24). Others simply stated that they had generally struggled on leaving, for example, 'the last night we were there, there was no sense of jubilation […] it was just very quiet, very subdued. [Leaving] affected me quite badly really […] I was absolutely lost' (KHP4), or that they had experienced negative reactions from others, for example, 'the driver who came to pick us up said "I will have to call head office to get the car disinfected after I drop you off"—that response I think will stay with me for a long time' (AP13).

The majority of participants did not receive follow-up information, though a few did receive information about sources of further support. While some stated that they would not have expected to receive any additional information, others felt that this would have been helpful, for example, 'I understand there is a lot happening right now…but I don't think there was enough support for us leaving' (AP23).

## DISCUSSION

This paper is the first in-depth analysis of the experiences of those who underwent supported isolation in the UK during the first wave of the COVID-19 pandemic. The findings therefore provide a unique insight into the way in which members of the public perceive supported isolation in the UK, and the factors that affect compliance and well-being in such settings. Given that supported isolation is once again required in the management of COVID-19 in the UK,[12 13 15] our findings should help facilitate optimised management of supported isolation procedures.

Despite some initial concerns, including confusion about what the process would involve and fears of infection, all willingly complied with the voluntary supported isolation process. People understood why it was necessary

and believed that doing so would protect themselves, their friends and family, and others in the UK; motivation for adherence was largely altruistic. Participants were overwhelmingly positive about their treatment by staff, communication from staff and overall management of the supported isolation process. This was fundamental to participants' willingness to comply with the restrictions of their liberty. Our findings are in line with systematic reviews carried out at the start of the pandemic,[7 22] as well as research into the management of other types of emergencies.[8 23] Crucially, participants believed their treatment by staff was legitimate, and they therefore chose to comply with supported isolation procedures; it is likely that compliance would have been much lower had staff attempted to enforce compliance.[23]

There were mixed views as to whether people in isolation experienced a connection with each other. However, almost all reported that others were helpful and friendly. Additionally, a number of people developed a shared identity with others; for example, they talked about everyone being in it together or going through the same experience. Those who did develop a shared identity often reported that this helped them to get through the process. This is as would be expected based on previous research which suggests that when people experience a sense of shared identity with others, this promotes adherence to protective measures, resilience and wellbeing.[8 9 11] While a sense of shared social identity arose spontaneously in some instances, participants emphasised that being able to communicate with others (eg, via chat groups) enhanced the social support that they experienced. Promoting virtual interaction between those undergoing supported isolation may be beneficial for strengthening shared identity, facilitating provision of social support and promoting resilience and well-being. Further research could examine how best to employ virtual methods (eg, WhatsApp groups, social media) to foster shared social identity and social support among those undergoing supported isolation, and the impact that this might have on experiences and behaviours during supported isolation. Participants also highlighted how important it was that they were able to easily keep in touch with friends and family during the supported isolation process.

While most participants reported either positive or neutral experiences during supported isolation, it was interesting to note that some reported negative experiences on leaving supported isolation. Findings suggest that it may be beneficial to prepare those undergoing supported isolation for possible psychosocial reactions they may experience on leaving supported isolation (eg, feeling anxious or overwhelmed), assist them with logistical aspects associated with leaving supported isolation (eg, organising travel home, contacting loved ones) and signpost them to sources of support. These would address many of the negative experiences on leaving supported isolation. However, some participants stated that they had struggled on leaving supported isolation but were not able to explain why that was the case. Further research should be carried out to better understand why some individuals may struggle on leaving supported isolation and improve support to these individuals.

The supported isolation carried out in January to February 2020 was designed to support those who were returning to the UK, and every effort was made to ensure that their experience was as positive as possible; as participants noted, staff could not do enough for them. Isolation in hotels is likely to be very different, with limited support from staff and an emphasis increasingly on enforcement rather than encouragement.[15] The reasons why people are travelling in the middle of a pandemic will also be different. The UK may find itself placing people into isolation who are more likely to experience distress such as those who are arriving to attend a funeral, are travelling due to a family crisis or who do not speak English. We must also not forget that, unlike travellers placed into facilities at Arrowe Park or Kents Hill, returning travellers will now be asked to pay £1500 each towards their isolation.

It is therefore critical that those responsible for implementing policies on isolation requirements take into account the recommendations presented here; failure to do so is likely to reduce adherence to isolation and risks serious long-term impact on those involved. Further research should explore travellers' experiences of undergoing supported isolation within one of the designated hotels. Due to the key differences (outlined above) between these hotels and the supported isolation reported in this paper, this should be compared with the experiences of those who underwent supported isolation at Arrowe Park or Kents Hill Park, and further our understanding of factors affecting compliance and well-being in supported isolation settings.

### Limitations

We have no information on those who did not participate, and it is possible that they differed on key variables. Of those who did, we reached thematic saturation within the sample. A second limitation is that only those who had a good understanding of English were interviewed. It is possible that the experience differed for those who were less able to understand English; indeed, this was alluded to in some comments made by participants. A final limitation is that this study was jointly run by King's College London and PHE, and PHE also assisted with the management of the supported isolation process. The team carrying out this research were not associated with the management of the supported isolation process, although did provide advice to the teams involved. It is therefore possible that participants were aware that PHE played a role in managing the supported isolation process.

## CONCLUSION AND RECOMMENDATIONS

Our findings, viewed in the context of the wider relevant published literature, generate several key recommendations that are particularly relevant given the requirement for some travellers to isolate in hotels. Specific recommendations are: (1) Prior to supported isolation, authorities should communicate with those affected about why isolation is necessary, how it will help to protect others and what the process will involve. Given that compliance is often motivated by altruism, emphasising how isolation will protect others is crucial. Such communication will also reduce concerns related to uncertainty about the isolation process. (2) Authorities should communicate effectively with those undergoing isolation throughout the process. Communication should be open and honest, and information should include protective actions people should take, why taking such actions is effective and how taking such actions protects oneself and others. (3) Enforcement of isolation should be avoided wherever possible. Given the large numbers of people who may be required to isolate at one time it will not be possible to enforce adherence; attempting to do so is likely to be perceived as illegitimate, thereby reducing adherence and risking serious long-term consequences for those involved. (4) It is likely to be helpful to facilitate and encourage development of shared identity among those undergoing supported isolation, via the formation of chat groups or other means of communication, that include staff managing the facilities. This type of shared social identity should encourage both adherence to supported isolation measures and improved resilience during the supported isolation process. (5) It is important to ensure that all essential supplies (such as food, exercise facilities, ability to communicate with those outside isolation) are provided and are suitable for the needs of the traveller. (6) Authorities should provide relevant information prior to leaving supported isolation to help people to prepare to return to their normal lives. Relevant information should cover the emotions that people might experience, and sources of further support that people can access if required. It may also be beneficial to include in this information any ongoing expectations around adherence to protective behaviours.

**Contributors** RA, HC, DW, NG, GJR, IO and SW conceived the study. HC, CR, DW and RA collected the data. HC and LG carried out data familiarisation and developed the coding framework. HC carried out the analysis and wrote the first draft of the manuscript. All authors contributed to the design and implementation of the study, and to the writing of the manuscript.

**Funding** This study was funded by the National Institute for Health Research Health Protection Research Unit (NIHR HPRU) in Emergency Preparedness and Response (grant number 200890), a partnership between Public Health England, King's College London and the University of East Anglia. DW, IO, CR and RA are supported by the NIHR HPRU in Behavioural Science and Evaluation, a partnership between Public Health England and the University of Bristol. CR is also supported by the NIHR HPRU in Emerging and Zoonotic Infections and NIHR HPRU in Gastrointestinal Infections.

**Disclaimer** The views expressed are those of the authors and not necessarily those of the NIHR, Public Health England or the Department of Health and Social Care.

**Competing interests** HC, DW, IO, CR and RA are current employees of Public Health England. GJR participates in the UK's Scientific Advisory Group for Emergencies and its subgroups.

**Patient consent for publication** Not required.

**Ethics approval** Ethical approval was obtained from the Public Health England Research Ethics Governance Group (approval number NR0187).

**Provenance and peer review** Not commissioned; externally peer reviewed.

**Data availability statement** Data are available upon reasonable request. Data are available on request from holly.carter@phe.gov.uk.

**ORCID iDs**
Holly Carter http://orcid.org/0000-0002-2084-7263
Simon Wessely http://orcid.org/0000-0002-6743-9929

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
