## [Reviewer comments · BMJ Open]

ARTICLE DETAILS

TITLE (PROVISIONAL)	Experiences of supported isolation in returning travellers during the early COVID-19 response: a qualitative interview study
AUTHORS	Carter, Holly; Weston, Dale; Greenberg, N; Oliver, Isabel; Robin, Charlotte; Rubin, GJ; Wessely, Simon; Gauntlett, Louis; Amlot, Richard

VERSION 1 – REVIEW

REVIEWER	Ye, Zhihong Zhejiang University School of Medicine Sir Run Run Shaw Hospital
REVIEW RETURNED	13-Mar-2021

GENERAL COMMENTS	Thank you for the opportunity to review your manuscript. As currently written this paper does not meet the standards required for publication in an international journal. I offer the following suggestions to help enhance the paper and its presentation to readers. Introduction The author did not distinguish between the definition of “quarantine” and “isolation”. According to this article, it should be “quarantine” rather than “isolation”. More information about quarantine would be helpful to the reader as it can be differently done depending on country and circumstance. The significance of research needs to be further enriched. The author mentions that the study is a mixed study; how does the research embody mixed research? Method The design section could be strengthened. What was the research question? And the aims and objectives of the research? In this paper, no methodology is discussed. What is sampling strategy? Why were interviews chosen as a data collection instrument? What advantage do interviews have given the topic of discussion? Did the data collection and analysis happen concurrently or where all interviews completed before data analysis was started? Could the authors provide some more information with regards to transferability and trustworthiness of qualitative analysis? Results Participants’ general demographic information was not provided.
---

	Discussion This section should consist of an in-depth analysis of the findings. This analysis seems very piecemeal and superficial. For example, “Of particular interest was our finding that some participants reported negative experiences on leaving supported isolation. It may therefore be beneficial to prepare participants for possible psychosocial reactions prior to them leaving supported isolation and signpost them to sources of support.”, the authors need more elaboration of leaving supported isolation may lead to negative experience. Areas for further research not addressed. The research is not innovative enough.
--	---

REVIEWER	Burton, Alexandra University College London, Division of Psychiatry
REVIEW RETURNED	14-Mar-2021

GENERAL COMMENTS	The topic of this paper is interesting and novel from an academic research perspective and has some important practical implications which hopefully have already fed into government decision making on supported quarantine. The paper could be strengthened by supplying further details of the analytical process and focusing/reorganising themes. Abstract 1. While the sample is very specific and novel, the objective seems very broad and may explain why so many themes and sub themes are identified. I wonder if it would make the paper stronger to provide further focus to the research objective(s) – i.e. ultimately it feels that the purpose of this study was to explore what might help or hinder compliance with supported isolation as part of the response to the COVID-19 pandemic, among people returning to the UK from Wuhan, China? The interview questions also appear to be trying to understand the impact on people’s wellbeing during this time? Strengths and limitations of the study 2. Page 4, line 35 - Remove or reword the first point from strengths and limitations - Supported isolation, or quarantine, is a key public health intervention that can be used to control the spread of COVID-19 – this is just summarising the rationale for the study rather than a strength or limitation of the study. Introduction 3. Page 5, line 26 - Could you state the number of countries or a bit more about why other countries are on the quarantine list, as “some other countries” sounds random and as if the number/rationale is unknown? Methods
--

4. Page 6, line 9 -This included an invitation to take part in a survey (findings reported elsewhere) -Can you add a reference to this work?

5. A bit more detail on consent is needed – e.g. was this written informed consent or verbal consent?

6. How was the interview guide developed? Was it piloted? Some of the questions listed seem to be designed to elicit yes/no responses (e.g. *Were you willing to undergo supported isolation*) Could the interview guide be provided as supplementary material?

7. Should the details of the sample be within the results section rather than methods? Also report mean/median age

8. The analytical process is not well described and seems to be missing some key steps and details:

a) How was the a priori thematic framework determined? Were the themes based on any pre-existing behavioural science theory/presumptions/constructs/evidence from previous pandemics?

b) The determination of only 12 themes within the framework sounds very restrictive. Were the data not coded more openly in the first instance (e.g. development of a coding framework rather than a thematic framework)?

c) How many of the 12 themes were a priori and how many were indeed identified during the analysis process? The themes are listed and then the authors state that “each passage was coded into one or more of the identified themes” which suggests that all themes were identified a priori? How were themes allowed to “emerge” from the data during this process?

d) Analysis was carried out “by hand” by the first author. I assume this means that software was not used to code/analyse or organise the data, however further description of how the analysis was conducted “by hand” is needed and how the framework analysis steps were taken (e.g. comparing and contrasting data) and tools (e.g. development of a matrix) used to organise the data.

e) Was there any checking in process/discussions about emerging findings within the team and if so can this be acknowledged as part of the analytical process? I would expect to see consideration of findings by multiple team members specifically when undertaking framework analysis

Results

9. Were no further characteristics collected about the sample beyond age, gender of participants and where they

quarantined? E.g. it would be interesting to know how many people were alone or with family/friends when they were quarantining

10. Some of the themes seem underdeveloped and descriptive with a lack of interpretation. There are an overwhelming number of themes and sub themes. It feels that a further stage of analysis would be helpful and would help strengthen the narrative if the concepts were grouped further into higher level themes to explore e.g. 1. factors that helped compliance; 2. factors that hindered/threatened compliance; 3. impact on wellbeing etc....

11. Could some of themes be reorganised/merged or removed? E.g.

a) The compliance theme says little more than the majority of people were willing to undergo isolation. It does not feel like a standalone theme and could be removed or reported quantitatively? I was left wondering what people's motivation for compliance or non-compliance was which is the purpose of further themes - the participant quote in the appendix suggests boredom was one reason for non-compliance which is explored in the feelings about isolation theme.

b) The areas for improvement theme again feels descriptive and could be reported quantitatively or removed and simply left for exploration in the discussion in response to other themes e.g. food quality is described in the management of supported isolation theme

12. When describing "themes" within themes these should be clearly labelled as "sub themes"

13. Most of the important information that should be contained within the manuscript is in the appendix:

a) None of the themes have supporting participant quotes. It is important to include quotes throughout the results section to show how participant accounts illustrate the themes and to aid in interpretation, rather than signposting the reader to the appendix. The appendix could then include more quotes per theme to illustrate them further

b) Similarly, the appendix introduces corresponding sub themes within themes. These sub themes are not explicitly mentioned in the manuscript (instead are referred to as themes and hidden within the text). A table introducing the themes and sub themes within the manuscript would be useful to help guide the reader, as well as independent consideration of the sub themes within the text.

Discussion

	14. Line 3 page 9 – Regarding the statement - participants were aware of what was going on around theme (sic), so the reports of very high compliance with supported isolation and other protective behaviours can be generalised to all those who were in quarantine..... Can you explain this further as it currently does not make sense? Why would knowing what was going on around you mean that results can be generalised to others? As this is qualitative research, generalisability of results is not the aim and you may want to instead consider transferability of your findings. However, ultimately, I don't think you can assume that the shared experience of quarantine means everyone experienced it in the same way as your participants did. Also given two thirds did not take part and you do not have information about non-responders I would advise that any reference to generalisability of findings be removed. Spelling correction 15. Line 3 page 8 – “Theme” should be “them” References 16. I would expect to see Richie and Spencer referenced for framework analysis: Ritchie, J. & Spencer, L. 1994. Qualitative data analysis for applied policy research" by Jane Ritchie and Liz Spencer in A. Bryman and R. G. Burgess [eds.] "Analyzing qualitative data", 1994, pp.173-194.
--	---

REVIEWER	Sheel, Meru Australian National University, National Centre for Epidemiology and Population Health
REVIEW RETURNED	31-Mar-2021

GENERAL COMMENTS	This is a generally well written useful description of supported/ managed isolation and quarantine (MIQ). However, considering MIQ has been in place for over a year, it is important for this article to reflect on current situation not just in the UK but also globally, especially considering the readership of the journal. Specifically:  1. Introduction – needs to reflect on managed isolation and quarantine in other countries. Many in the Asia-Pacific region have been doing it for over a year now; and that context especially countries with similar social and economic contexts like Australia and New Zealand. 2. Introduction - Important to describe the MIQ – for eg what were the participants allowed to do, did they have access to fresh air? Were they allowed to leave their rooms? Mask use? Food provision? Exercise etc these are important variables that will influence people's experience so important to describe these. Any differences in both facilities surveyed? 3. Line 14 – participant age in results 4. Strength: within one month of undergoing MIQ 5. Results (line 39): why do participants need to practice protective
---

	measures in isolation – relates back to context? Were they allowed out or were they in shared facility? Were they moving? 6. Line 36 – how were people in isolation able to communicate with others? 7. The results are surprisingly positively skewed which are interesting to read. The role of government agencies in this study may have a role, but has been rightly so been acknowledged by the authors as a limitation but does not discount the findings of the study.
--	---

VERSION 1 – AUTHOR RESPONSE

Reviewer: 1

Dr. Zhihong Ye, Zhejiang University School of Medicine Sir Run Run Shaw Hospital

Comments to the Author:

Thank you for the opportunity to review your manuscript. As currently written this paper does not meet the standards required for publication in an international journal. I offer the following suggestions to help enhance the paper and its presentation to readers.

Introduction

The author did not distinguish between the definition of “quarantine” and “isolation”. According to this article, it should be “quarantine” rather than “isolation”.

“Supported isolation” and “quarantine” typically describe the same type of managed isolation process (i.e. isolation outside of one’s own home, and so distinct from self-isolation). In the UK, the process is typically referred to as supported isolation, rather than quarantine, and so we have referred to it as supported isolation throughout. We have now clarified this in the introduction on page 4: “In order to be repatriated all had to agree to undergo 14 days of ‘supported isolation’. In some countries and contexts this type of supported isolation is known as quarantine; however, it is typically referred to as supported isolation in the UK, and so will be referred to as supported isolation in the current study.”

More information about quarantine would be helpful to the reader as it can be differently done depending on country and circumstance.

We agree that information about how the process was carried out will be beneficial to the reader. We have therefore now added the following sentences to the introduction on page 4, to explain what supported isolation involved in this instance: “On arrival at the supported isolation facility, individuals were provided with their own rooms which were fully furnished and had basic cooking, washing and living facilities [3]. Individuals were encouraged to stay in their rooms as much as possible (though this was not mandatory) and could access anything they needed by phoning staff or using an online system; if they did

need to leave their rooms they were encouraged to follow hand hygiene guidance and wear a face mask. Individuals also had access to a team of medical staff who closely monitored their condition, including regular testing and symptom checking [3]. There was phone and internet access to enable them to communicate with others both inside and outside the supported isolation facility.”

The significance of research needs to be further enriched.

We have now provided further detail about the significance of the research, particularly in light of the fact that supported isolation is now required in the UK for travellers from many different countries, and have added the following text to the introduction on page 5: “To our knowledge, this is the first research conducted with individuals during and immediately following their supported isolation in this country. With supported isolation now being required for people travelling to the UK from a number of countries, the findings presented here will be invaluable in understanding public experiences of supported isolation and informing optimised management in these settings.”

The author mentions that the study is a mixed study; how does the research embody mixed research?

We have now clarified that this was a mixed methods study involving semi-structured interviews (reported in the current manuscript) and surveys (reported elsewhere). We have included the following text in the introduction on page 5, including a reference directing the reader to findings from the survey data: “We carried out a rapid mixed-methods study in which we: 1) interviewed individuals who underwent supported isolation at Arrowe Park Hospital and Kents Hill Park conference centre (findings reported here); 2) surveyed those who underwent supported isolation at two time points (immediately after supported isolation and three months after supported isolation) (findings reported elsewhere; Carter et al., in prep).”

Method

The design section could be strengthened.

We have now added the following information to clarify the method used, and why this method was chosen, and therefore strengthen the design section: “The decision was taken to carry out semi-structured interviews (alongside surveys, reported elsewhere; Carter et al., in prep) in order to generate a more in-depth understanding of participants’ perceptions and experiences during supported isolation than could be obtained using surveys alone.”

What was the research question? And the aims and objectives of the research?

We have now added an ‘Aims’ sub-section to the introduction on page 5, to specifically outline the aims of the research: “This study had two aims: 1) to understand the experiences and perceptions of those who underwent supported isolation, particularly in relation to factors that were associated with improved compliance and wellbeing; 2) to inform the development of recommendations for the management of similar supported isolation procedures.”

In this paper, no methodology is discussed.

We have now considerably expanded the original method section, including adding information to the design, participants, procedure and analysis sections. In particular, we have added substantial detail to our description of the way in which analysis was carried out, as well as clarifying the method used and why this method was chosen. We feel that the method section is considerably strengthened as a result.

What is sampling strategy?

We have now added the following information to the participants section on pages 5 – 6, to clarify the sampling strategy: “The day before leaving supported isolation, all those in the supported isolation facilities were provided with an information sheet about the study by a member of staff at the facility. This included an invitation to take part in a survey (findings reported elsewhere; Carter et al., in prep), as well as the opportunity to take part in an interview. Thus, voluntary response sampling was used, whereby all those who underwent supported isolation were given the opportunity to take part in both the survey (reported elsewhere; Carter et al., in prep) and an interview, and the sample consisted of those who chose to opt-in to the study. To opt-in to the interview part of the study, participants were asked to provide an email address on leaving supported isolation to enable the research team to follow up and arrange the interview.”

Why were interviews chosen as a data collection instrument? What advantage do interviews have given the topic of discussion?

We have now added the following text to the design section on page 5, to clarify why we chose to carry out interviews: “The decision was taken to carry out semi-structured interviews (alongside surveys, reported elsewhere; Carter et al., in prep) in order to generate a more in-depth understanding of participants’ perceptions and experiences during supported isolation than could be obtained using surveys alone.”

Did the data collection and analysis happen concurrently or where all interviews completed before data analysis was started?

All interviews were completed before data analysis was started, and we have now added the following sentence to the analysis section on page 6 to clarify this: “All interviews were completed before beginning data analysis, at which point a framework approach was used to analyse the data [19].”

Could the authors provide some more information with regards to transferability and trustworthiness of qualitative analysis?

Whilst qualitative data are not by their nature generalisable, they can be transferable to similar contexts. In this instance, the data collected from those who had undergone supported isolation on their return to the UK is transferable to similar supported isolation contexts now required in the UK, such as quarantine hotels. In terms of trustworthiness or reliability of the data, this can be achieved by ensuring that data

saturation has been reached (as it has been in the current study), following a rigorous and robust analytical approach (in this case the framework approach), and involving more than one researcher in data analysis where possible (in the current study, a second researcher was involved in data familiarisation and the development of a coding framework). As noted above, we have now considerably expanded the analysis section to include more detail about the analytical approach that we followed, and the steps involved in that analysis. We therefore feel that we have now provided more detail about the transferability and trustworthiness of the qualitative data collected in the current study.

Results

Participants' general demographic information was not provided.

We have now added a demographics sub-section to the results section on page 7: "Half of the participants (n = 13) were male and half (n = 13) were female. Participants ranged in age from 22 to 78 (mean = 43.2 years). The majority of participants were British nationals (n = 22), with a small number of Chinese nationals (n = 3) and one person who selected 'Other' as their nationality. Similarly, the majority of participants were White British (n = 17), or Chinese (n = 7), with one person being White Irish, and another being Black British. Most participants were educated to degree level or above (n = 17), with a smaller number being educated to higher secondary level (n = 8), and one being educated to primary or lower secondary level. The majority of participants were employed either full time (n = 14) or part time (n = 4). A small number were retired (n = 4), unemployed (n = 2) or self-employed (n = 1), with one participant specifying that they were due to start work following their isolation.

Participants were asked what their reason was for being in Wuhan during the COVID-19 outbreak, and most stated that they were either living there (n = 6), visiting family or friends (n = 8), or on holiday (n = 5). A smaller number were there on a business trip (n = 2), with one participant having been deployed as part of the FCO response. A small number stated that they had not been in Wuhan and were isolating on their return from other affected areas, including Hubei province (n = 2) and the Diamond Princess cruise ship (n = 2). The majority of participants were travelling either with family (n = 11) or on their own (n = 10), with a small number traveling with others they had no relationship with (n = 5). The majority of participants did not share a room (n = 15). Of those that did (n = 11), most shared with family (n = 7) or friends (n = 1), with only a small number sharing with people they didn't know (n = 3)."

Discussion

This section should consist of an in-depth analysis of the findings. This analysis seems very piecemeal and superficial. For example, "Of particular interest was our finding that some participants reported negative experiences on leaving supported isolation. It may therefore be beneficial to prepare participants for possible psychosocial reactions prior to them leaving supported isolation and signpost them to sources of support.", the authors need more elaboration of leaving supported isolation may lead to negative experience.

We have now expanded the discussion to include more in-depth analysis, as the reviewer suggests. In particular, we have expanded the point around negative feelings on leaving supported isolation: “While most participants reported either positive or neutral experiences during supported isolation, it was interesting to note that some participants reported negative experiences on leaving supported isolation. Findings suggest that it may be beneficial to prepare those undergoing supported isolation for possible psychosocial reactions they may experience upon leaving supported isolation (e.g. feeling anxious or overwhelmed), assist them with logistical aspects associated with leaving supported isolation (e.g. organising travel home, contacting loved ones), and signpost them to sources of support. These would address many of the negative experiences upon leaving supported isolation. However, some participants stated that they had struggled on leaving supported isolation but were not able to explain why that was the case. Further research should be carried out to better understand why some individuals may struggle on leaving supported isolation and improve support to these individuals.”

Areas for further research not addressed.

We have now added three areas for further research to the discussion: “Further research could examine how best to employ virtual methods (e.g. WhatsApp groups, social media) to foster shared social identity and social support amongst those undergoing supported isolation, and the impact that this might have on experiences and behaviours during supported isolation.” “Further research should be carried out to better understand why some individuals may struggle on leaving supported isolation, and to improve support to these individuals.” “Further research should explore travellers’ experiences of undergoing supported isolation within one of the designated hotels.”

The research is not innovative enough.

Whilst we respect the reviewer’s opinion, we believe that the research is innovative in being the first study to carry out in-depth qualitative analysis to understand experiences and perceptions of supported isolation in the UK; indeed, this is reflected in the comments of Reviewer 2. We have added the following text to the discussion, to emphasise the unique value of the study: “This paper is the first in-depth analysis of the experiences of those who underwent supported isolation in the UK during the first wave of the COVID-19 pandemic. The findings therefore provide a unique insight into the way in which members of the public perceive supported isolation in the UK, and the factors that affect compliance and wellbeing in such settings.”

Reviewer: 2

Miss Alexandra Burton, University College London

Comments to the Author:

The topic of this paper is interesting and novel from an academic research perspective and has some important practical implications which hopefully have already fed into government decision making on supported quarantine. The paper could be strengthened by supplying further details of the analytical

process and focusing/reorganising themes. Please refer to the attached word document for specific feedback

The topic of this paper is interesting and novel from an academic research perspective and has some important practical implications which hopefully have already fed into government decision making on supported quarantine. The paper could be strengthened by supplying further details of the analytical process and focusing/reorganising themes.

We thank the reviewer for their feedback. We have now provided further information about the analytical process used and have reorganised the way themes are presented (details provided under specific comments, below) and we feel that the paper has been strengthened considerably as a result.

Abstract

1. While the sample is very specific and novel, the objective seems very broad and may explain why so many themes and sub themes are identified. I wonder if it would make the paper stronger to provide further focus to the research objective(s) – i.e. ultimately it feels that the purpose of this study was to explore what might help or hinder compliance with supported isolation as part of the response to the COVID-19 pandemic, among people returning to the UK from Wuhan, China? The interview questions also appear to be trying to understand the impact on people's wellbeing during this time?

We have now made the objectives section of the abstract more specific, in line with the reviewer's suggestion: "Objectives: 1) to understand the experiences and perceptions of those who underwent supported isolation, particularly in relation to factors that were associated with improved compliance and wellbeing; 2) to inform the development of recommendations for the management of similar supported isolation procedures."

Strengths and limitations of the study

2. Page 4, line 35 - Remove or reword the first point from strengths and limitations - Supported isolation, or quarantine, is a key public health intervention that can be used to control the spread of COVID-19 – this is just summarising the rationale for the study rather than a strength or limitation of the study.

We have now removed this point.

Introduction

3. Page 5, line 26 - Could you state the number of countries or a bit more about why other countries are on the quarantine list, as "some other countries" sounds random and as if the number/rationale is unknown?

We have now expanded this section to provide more information about why some countries are on the quarantine list: "From 15th February 2021 those travelling to the UK from 'red list' countries (countries which have higher prevalence of new COVID-19 variants) [12] have been required to isolate in hotels for 10 days [13]. Countries on the 'red list' are continually reviewed and updated, but as of 9th April 2021 there were 39 countries on the list."

Methods

4. Page 6, line 9 -This included an invitation to take part in a survey (findings reported elsewhere) - Can you add a reference to this work?

We have now added a reference for the survey work.

5. A bit more detail on consent is needed – e.g. was this written informed consent or verbal consent?

We have now added the following information to the 'Procedure' section to clarify the way in which consent was provided: "Prior to taking part in an interview, participants completed a written consent form. They also provided verbal consent at the start of the interview."

6. How was the interview guide developed? Was it piloted? Some of the questions listed seem to be designed to elicit yes/no responses (e.g. Were you willing to undergo supported isolation) Could the interview guide be provided as supplementary material?

The interview guide was developed by the research team; given the timescales involved it was unfortunately not possible to pilot test the schedule. The interview schedule is now provided as supplementary material, as suggested (Appendix 2).

7. Should the details of the sample be within the results section rather than methods? Also report mean/median age

We have now expanded the details of the sample (see response to Reviewer 1 comment relating to demographics, above), including adding mean age, and have moved this to a demographics sub-section within the results.

8. The analytical process is not well described and seems to be missing some key steps and details:

a) How was the a priori thematic framework determined? Were the themes based on any pre-existing behavioural science theory/presumptions/constructs/evidence from previous pandemics?

The a priori framework was determined based on areas of policy interest, as well as aspects that have been shown to be important for compliance and wellbeing during similar types of incidents (e.g. risk perception in relation to COVID-19, perceived management of supported isolation, communication from staff). We have added the following text to the analysis section to explain how the a priori framework was developed: "After familiarisation with the data, an initial coding framework was developed based largely on a priori areas of interest in line with the research aims, and specifically included factors that have been shown during previous incidents to be related to compliance and wellbeing. At this stage, themes were also allowed to emerge from the data. The initial coding framework was intentionally broad, to ensure that areas of interest were not missed, and contained a total of 76 categories, within 22 major themes."

b) The determination of only 12 themes within the framework sounds very restrictive. Were the data not coded more openly in the first instance (e.g. development of a coding framework rather than a thematic framework)?

We thank the reviewer for highlighting the need to be more specific in the way in which we coded data. The description of 12 themes refers to the final thematic framework (though we have also now revised some of the themes, also in line with the reviewer's suggestion). We have added the following paragraph to the Analysis section in order to clarify all the stages of development of both the coding framework and the analytical framework: "After familiarisation with the data, an initial coding framework was developed based largely on a priori areas of interest in line with the research aims, and specifically included factors that have been shown during previous incidents to be related to compliance and wellbeing. At this stage, themes were also allowed to emerge from the data. The initial coding framework was intentionally broad, to ensure that areas of interest were not missed, and contained a total of 76 categories, within 22 major themes. The initial framework was discussed with a second researcher, who had also familiarised themselves with the data, and then applied to a small number of transcripts. The initial coding framework was then refined into an analytical framework, in which codes were grouped together into overarching themes. This resulted in 6 key themes, and 7 sub-themes: factors affecting compliance with supported isolation (two sub-themes: factors promoting compliance; factors threatening compliance); risk perceptions around catching COVID-19 (two sub-themes: low perceived risk; high perceived risk); management of supported isolation (three sub-themes: operational management; treatment by staff; communication from staff); communication with those outside of supported isolation facilities; relationship with others within supported isolation; feelings on leaving supported isolation."

c) How many of the 12 themes were a priori and how many were indeed identified during

the analysis process? The themes are listed and then the authors state that “each passage was coded into one or more of the identified themes” which suggests that all themes were identified a priori? How were themes allowed to “emerge” from the data during this process?

As described in our response to comment 8b (above) we have now updated the Analysis section to clarify how we developed both the coding and analytical frameworks. We hope this clarifies that the initial coding framework was developed based both on a priori issues of interest in line with the research aims, but also as a result of themes which emerged during familiarisation.

d) Analysis was carried out “by hand” by the first author. I assume this means that software was not used to code/analyse or organise the data, however further description of how the analysis was conducted “by hand” is needed and how the framework analysis steps were taken (e.g. comparing and contrasting data) and tools (e.g. development of a matrix) used to organise the data.

We have now updated the Analysis section to explain in more detail how analysis was carried out: “Application of the analytical framework was carried out by hand by the first author, with each passage in the data being coded into one or more of the identified themes. A spreadsheet was used to generate a matrix into which relevant data (e.g. passages of interest relating to each theme) were organised thematically. This enabled data to be compared and contrasted within and between themes and facilitated more in-depth interpretation.”

e) Was there any checking in process/discussions about emerging findings within the team and if so can this be acknowledged as part of the analytical process? I would expect to see consideration of findings by multiple team members specifically when undertaking framework analysis

A second researcher was involved in familiarisation, development of the initial coding framework, and refinement of the analytical framework (following application to a small number of transcripts). We have updated the analysis section to reflect this: “The initial framework was discussed with a second researcher, who had also familiarised themselves with the data, and then applied to a small number of transcripts. The initial coding framework was then refined into an analytical framework, in which codes were grouped together into overarching themes.”

Results

9. Were no further characteristics collected about the sample beyond age, gender of participants

and where they quarantined? E.g. it would be interesting to know how many people were alone or with family/friends when they were quarantining

We have now added the following information to the 'Results' section: "Half of the participants (n = 13) were male and half (n = 13) were female. Participants ranged in age from 22 to 78 (mean = 43.2 years). The majority of participants were British nationals (n = 22), with a small number of Chinese nationals (n = 3) and one person who selected 'Other' as their nationality. Similarly, the majority of participants were White British (n = 17), or Chinese (n = 7), with one person being White Irish, and another being Black British. Most participants were educated to degree level or above (n = 17), with a smaller number being educated to higher secondary level (n = 8), and one being educated to primary or lower secondary level. The majority of participants were employed either full time (n = 14) or part time (n = 4). A small number were retired (n = 4), unemployed (n = 2) or self-employed (n = 1), with one participant specifying that they were due to start work following their isolation.

Participants were asked what their reason was for being in Wuhan during the COVID-19 outbreak, and most stated that they were either living there (n = 6), visiting family or friends (n = 8), or on holiday (n = 5). A smaller number were there on a business trip (n = 2), with one participant having been deployed as part of the FCO response. A small number stated that they had not been in Wuhan and were isolating on their return from other affected areas, including Hubei province (n = 2) and the Diamond Princess cruise ship (n = 2). The majority of participants were travelling either with family (n = 11) or on their own (n = 10), with a small number traveling with others they had no relationship with (n = 5). The majority of participants did not share a room (n = 15). Of those that did (n = 11), most shared with family (n = 7) or friends (n = 1), with only a small number sharing with people they didn't know (n = 3)."

10. Some of the themes seem underdeveloped and descriptive with a lack of interpretation. There are an overwhelming number of themes and sub themes. It feels that a further stage of analysis would be helpful and would help strengthen the narrative if the concepts were grouped further into higher level themes to explore e.g. 1. factors that helped compliance; 2. factors that hindered/threatened compliance; 3. impact on wellbeing etc....

We agree that there are a large number of themes, and that some themes could be re-organised. We have therefore re-organised several themes, including the addition of sub-themes to strengthen the narrative. Specifically, we have amalgamated the 'Compliance' and 'Feelings about supported isolation' themes into one 'Factors affecting compliance' theme, with two sub-themes: 'Factors promoting compliance'; 'Factors threatening compliance'. We have also created a broad 'Management of supported isolation' theme, which includes three sub-themes: 'Operational management'; 'Treatment by staff' and 'Communication from staff'. We have also removed the 'Areas for improvement' theme, incorporating suggestions for improvement into appropriate themes. We feel that broadening the themes, and

developing sub-themes, has enabled us to strengthen the narrative of the paper, and focus on particular areas of policy and research interest.

11. Could some of themes be reorganised/merged or removed? E.g.

a) The compliance theme says little more than the majority of people were willing to undergo isolation. It does not feel like a standalone theme and could be removed or reported quantitatively? I was left wondering what people's motivation for compliance or non-compliance was which is the purpose of further themes - the participant quote in the appendix suggests boredom was one reason for non-compliance which is explored in the feelings about isolation theme.

We agree that there is substantial overlap between the compliance theme and the feelings about support isolation them. We have therefore now amalgamated the two themes into one overall compliance theme, but also created two sub-themes within the compliance theme (factors promoting compliance, and factors threatening compliance) as the reviewer suggests. We feel that this has really helped to draw together the factors that may be associated with improved, or reduced, compliance.

b) The areas for improvement theme again feels descriptive and could be reported quantitatively or removed and simply left for exploration in the discussion in response to other themes e.g. food quality is described in the management of supported isolation theme

On reflection, we agree with the reviewer that several of the areas for improvement (including points around the importance of pre-supported isolation information, and improved food provision) have already been covered in other themes. Additionally, the point raised around the importance of access to exercise facilities and outside space related to operational management of supported isolation. All suggested areas for improvement have now been described under other themes, and the areas for improvement theme has been removed.

12. When describing "themes" within themes these should be clearly labelled as "sub themes"

We agree that the labelling of sub-themes within the text was not consistent. We have therefore now added the following text to the analysis section, to outline the themes and sub-themes reported: "This resulted in 6 key themes, and 7 sub-themes: factors affecting compliance with supported isolation (two sub-themes: factors promoting compliance; factors threatening compliance); risk perceptions around catching COVID-19 (two sub-themes: low perceived risk; high perceived risk); management of supported isolation (three sub-themes: operational management; treatment by staff; communication from staff); communication with those outside of supported isolation facilities; relationship with others within

supported isolation; feelings on leaving supported isolation". We have also clearly and consistently labelled the themes and sub-themes in the results section.

13. Most of the important information that should be contained within the manuscript is in the appendix:

a) None of the themes have supporting participant quotes. It is important to include quotes throughout the results section to show how participant accounts illustrate the themes and to aid in interpretation, rather than signposting the reader to the appendix. The appendix could then include more quotes per theme to illustrate them further

As suggested by the reviewer, we have now moved supporting quotes from Appendix 2 into the text of the results section. We agree that this has improved the narrative of the section and should help with interpretation of findings.

b) Similarly, the appendix introduces corresponding sub themes within themes. These sub themes are not explicitly mentioned in the manuscript (instead are referred to as themes and hidden within the text). A table introducing the themes and sub themes within the manuscript would be useful to help guide the reader, as well as independent consideration of the sub themes within the text.

As noted under comment 12, above, we have now specifically outlined the themes and sub-themes by describing these in the analysis section, and by clearly and consistently labelling themes and sub-themes throughout the results section. We have also added a table (Table 1) to introduce the themes and sub-themes, as suggested by the reviewer.

Discussion

14. Line 3 page 9 – Regarding the statement - participants were aware of what was going on around theme (sic), so the reports of very high compliance with supported isolation and other protective behaviours can be generalised to all those who were in quarantine..... Can you explain this further as it currently does not make sense? Why would knowing what was going on around you mean that results can be generalised to others? As this is qualitative research, generalisability of results is not the aim and you may want to instead consider transferability of your findings. However, ultimately, I don't think you can assume that the shared experience of quarantine means everyone experienced

it in the same way as your participants did. Also given two thirds did not take part and you do not have information about non-responders I would advise that any reference to generalisability of findings be removed.

As suggested by the reviewer, we have now removed the statement around the extent to which findings can be generalised.

Spelling correction

15. Line 3 page 8 – “Theme” should be “them”

This has now been corrected.

References

16. I would expect to see Richie and Spencer referenced for framework analysis: Ritchie, J. & Spencer, L. 1994. Qualitative data analysis for applied policy research" by Jane Ritchie and Liz Spencer in A. Bryman and R. G. Burgess [eds.] "Analyzing qualitative data", 1994, pp.173-194.

As the reviewer suggests, we have now added the above reference to the description of framework analysis: "All interviews were completed before beginning data analysis, at which point a framework approach was used to analyse the data (Spencer & Ritchie, 1994)."

Reviewer: 3

Dr. Meru Sheel, Australian National University

Comments to the Author:

This is a generally well written useful description of supported/ managed isolation and quarantine (MIQ). However, considering MIQ has been in place for over a year, it is important for this article to reflect on current situation not just in the UK but also globally, especially considering the readership of the journal.

Specifically:

1. Introduction – needs to reflect on managed isolation and quarantine in other countries. Many in the Asia-Pacific region have been doing it for over a year now; and that context especially countries with similar social and economic contexts like Australia and New Zealand.

We have now provided further information to reflect the quarantine context in other countries: "Many countries, including China [4], Vietnam [5], and Singapore [6] have had supported isolation policies in place in response to COVID-19 for over a year, for a variety of situations including international travel.

However, supported isolation for returning travellers had, to our knowledge, never been used before within the UK.”

2. Introduction - Important to describe the MIQ – for eg what were the participants allowed to do, did they have access to fresh air? Were they allowed to leave their rooms? Mask use? Food provision? Exercise etc these are important variables that will influence people’s experience so important to describe these. Any differences in both facilities surveyed?

We have now added the following information to the introduction, to describe the circumstances within the supported isolation facilities: “On arrival at the supported isolation facility, individuals were provided with their own rooms which were fully furnished and had basic cooking, washing and living facilities [3]. Individuals were encouraged to stay in their rooms as much as possible (though this was not mandatory) and could access anything they needed by phoning staff or using an online system; if they did need to leave their rooms they were encouraged to follow hand hygiene guidance and wear a face mask. Individuals also had access to a team of medical staff who closely monitored their condition, including regular testing and symptom checking [3]. There was phone and internet access to enable them to communicate with others both inside and outside the supported isolation facility.”

3. Line 14 – participant age in results

Participant age, along with other demographic information, has now been moved to a ‘Demographics’ sub-section within the results section.

4. Strength: within one month of undergoing MIQ

We have now added the following bullet point as an additional strength of the study: “Interviews were carried out within one month of participants leaving supported isolation.”

5. Results (line 39): why do participants need to practice protective measures in isolation – relates back to context? Were they allowed out or were they in shared facility? Were they moving?

As noted under comment 2, above, we have now expanded the context to explain the conditions within the supported isolation facilities. This includes an explanation that, while people were encouraged to stay in their rooms as much as possible, this was not mandatory.

6. Line 36 – how were people in isolation able to communicate with others?

We have now added the following text to the introduction to explain how people in supported isolation were able to communicate with others: “There was phone and internet access to enable them to communicate with others both inside and outside the supported isolation facility.”

7. The results are surprisingly positively skewed which are interesting to read. The role of government agencies in this study may have a role, but has been rightly so been acknowledged by the authors as a limitation but does not discount the findings of the study.

We agree that it was surprising how positive the results were from this study, and that it is important to acknowledge that PHE (though not the research team involved in this study) had a role in setting up and running the supported isolation. As the reviewer notes, this is acknowledged as a potential limitation of the study. We have also expanded the limitation slightly, to further clarify the role of PHE more broadly, and the research team specifically: "A final limitation is that this study was jointly run by King's College London and Public Health England, and Public Health England also assisted with the management of the supported isolation process. The team carrying out this research were not associated with the management of the supported isolation process, although did provide advice to the teams involved. It is therefore possible that participants were aware that PHE played a role in managing the supported isolation process."

VERSION 2 – REVIEW

REVIEWER	Ye, Zhihong Zhejiang University School of Medicine Sir Run Run Shaw Hospital
REVIEW RETURNED	22-May-2021

GENERAL COMMENTS	Thank you for the opportunity to review the revised manuscript. The authors are to be commended for undertaking the significant revisions that were recommended. The manuscript now reads smoothly, is well-organized, and there is sufficient detail about the methods. Relevant information was added to the Introduction, and the conclusions reached in the Discussion section are well-described and follow from the findings. I do not have further suggestions for revision. This is a timely and interesting study that will be of interest to many readers.
--

REVIEWER	Burton, Alexandra University College London, Division of Psychiatry
REVIEW RETURNED	02-Jun-2021

GENERAL COMMENTS	Thank you for the revised manuscript. This addresses my previous comments. Just a few minor comments that need to be addressed: I would remove the list of themes and sub themes from your analysis section as it is repetitive and should only be in the results section - it is enough to refer the reader to table 1 in your analysis section after you state how many themes/sub themes were identified in your analysis. In your results section I would report your demographics and demographics table first and then your Table of themes so that the more detailed description of your themes immediately follows the table There is a random title in the results section: "Focus group discussions" This needs deleting?
---

VERSION 2 – AUTHOR RESPONSE

Reviewer: 1

Dr. Zhihong Ye, Zhejiang University School of Medicine Sir Run Run Shaw Hospital

Comments to the Author:

Thank you for the opportunity to review the revised manuscript. The authors are to be commended for undertaking the significant revisions that were recommended. The manuscript now reads smoothly, is well-organized, and there is sufficient detail about the methods. Relevant information was added to the Introduction, and the conclusions reached in the Discussion section are well-described and follow from the findings. I do not have further suggestions for revision. This is a timely and interesting study that will be of interest to many readers.

We thank Reviewer 1 for their feedback, and we are happy that all comments have been addressed and that the manuscript will be of interest to readers. We agree that the manuscript is considerably stronger as a result of the revisions undertaken.

Reviewer: 2

Miss Alexandra Burton, University College London

Comments to the Author:

Thank you for the revised manuscript. This addresses my previous comments. Just a few minor comments that need to be addressed:

I would remove the list of themes and sub themes from your analysis section as it is repetitive and should only be in the results section - it is enough to refer the reader to table 1 in your analysis section after you state how many themes/sub themes were identified in your analysis.

We agree that including this information in the analysis section is repetitive, and so as suggested we have now removed the list of themes and sub themes from the analysis section and just referred readers to Table 1.

In your results section I would report your demographics and demographics table first and then your Table of themes so that the more detailed description of your themes immediately follows the table

We agree that it is more helpful for the reader to have the detailed description of themes immediately following the table, and we have therefore moved Table 1 so it now appears after the demographics.

There is a random title in the results section: "Focus group discussions" This needs deleting?

We have now removed this title from the Results section.